# 24-nt reproductive phasiRNAs are broadly present in angiosperms

Rui Xia [1,2,3,4], Chengjie Chen[1,2,3], Suresh Pokhrel[4,5], Wuqiang Ma[1,2,3], Kun Huang[6], Parth Patel[6], Fuxi Wang[7], Jing Xu[3], Zhongchi Liu [7], Jianguo Li[1,2,3] & Blake C. Meyers [4,5]

Small RNAs are key regulators in plant growth and development. One subclass, phased siRNAs (phasiRNAs) require a trigger microRNA for their biogenesis. In grasses, two pathways yield abundant phasiRNAs during anther development; miR2275 triggers one class, 24-nt phasiRNAs, coincident with meiosis, while a second class of 21-nt phasiRNAs are present in premeiotic anthers. Here we report that the 24-nt phasiRNA pathway is widely present in flowering plants, indicating that 24-nt reproductive phasiRNAs likely originated with the evolutionary emergence of anthers. Deep comparative genomic analyses demonstrated that this miR2275/24-nt phasiRNA pathway is widely present in eudicots plants, however, it is absent in legumes and in the model plant Arabidopsis, demonstrating a dynamic evolutionary history of this pathway. In Solanaceae species, 24-nt phasiRNAs were observed, but the miR2275 trigger is missing and some loci displaying 12-nt phasing. Both the miR2275-triggered and Solanaceae 24-nt phasiRNAs are enriched in meiotic stages, implicating these phasiRNAs in anther and/or pollen development, a spatiotemporal pattern consistent in all angiosperm lineages that deploy them.

[1] State Key Laboratory for Conservation and Utilization of Subtropical Agro-Bioresources, South China Agricultural University, 510640 Guangzhou, Guangdong, China. [2] Key Laboratory of Biology and Germplasm Enhancement of Horticultural Crops in South China, Ministry of Agriculture, South China Agricultural University, 510640 Guangzhou, Guangdong, China. [3] Guangdong Litchi Engineering Research Center, College of Horticulture, South China Agricultural University, 510640 Guangzhou, Guangdong, China. [4] Donald Danforth Plant Science Center, St. Louis, MO 63132, USA. [5] Division of Plant Sciences, University of Missouri-Columbia, Columbia, MO 65211, USA. [6] Delaware Biotechnology Institute and Department of Plant and Soil Sciences, University of Delaware, Newark, DE 19711, USA. [7] Department of Cell Biology and Molecular Genetics, University of Maryland, College Park, MD 20742, USA. Correspondence and requests for materials should be addressed to R.X. (email: rxia@scau.edu.cn) or to B.C.M. (email: bmeyers@danforthcenter.org)

Phased secondary siRNAs or "phasiRNAs" in plants play key regulatory roles in growth, development, stress responses, and of importance to this work, in reproduction[1–4]. They are typically triggered by a 22-nt microRNA (miRNA) directing Argonaute-mediated slicing of the primary precursor ("PHAS") transcript, leading to recruitment of enzymes that make the sliced mRNA double-stranded and then sequentially process it by a Dicer-like (DCL) protein into 21- or 24-nt siRNAs. In rice and maize, hundreds of loci on all chromosomes yield abundant 21-nt phasiRNAs in premeiotic anthers[4–6] and 24-nt phasiRNAs enriched in meiotic-stage anthers[4,5]. Perturbation of 21-PHAS loci underlies agronomically important, photoperiod-sensitive male sterility in rice[7], while disruption of 24-nt phasiRNA production yields conditional male sterility in maize[8]. The evolutionary origins and phylogenetic distribution of these small RNA (sRNA) pathways have not been described.

The 24-nt phasiRNAs are triggered by miR2275, thus far characterized only in monocots[4], and then produced by a monocot-specific DCL protein, DCL5[4,9]. The absence of miR2275 and DCL5 in genomes of well-studied eudicot families (Brassicaceae, Fabaceae, Solanaceae) led us and others to hypothesize the restriction of this 24-nt phasiRNA pathway to the monocots[4]. Here, we report that this pathway is widely present in eudicots, consistent with an important role in flowering plant reproductive biology.

## Results

**Discovery in litchi of the pathway generating 24-nt phasiRNAs.** During an analysis of litchi (*Litchi chinenses* Sonn.) sRNA data generated previously[10], we unexpectedly identified three miR2275 stem–loop precursors yielding six canonical miRNA duplexes (two abundantly expressed example precursors shown in Fig. 1a). Two precursors encode polycistronic copies of miR2275 (Fig. 1a, b, Supplementary Fig. 1A and B). Mature miR2275 accumulated substantially in flowers (Fig. 1c). The only known role of miR2275 is to trigger 24-nt reproductive phasiRNAs; therefore, we sought these "24-PHAS" loci in litchi. We identified >178 24-PHAS loci; 113 had a target motif for miR2275, the only conserved sequence element in these loci (Fig. 1d, Supplementary Data 1). miR2275-directed cleavage was confirmed by PARE data, initiating the registry of the sRNA phasing (exemplified in Supplementary Fig. 1C), for floral phasiRNAs (Fig. 1e). These results were strikingly similar to our prior observations in maize[4], yet litchi, an eudicot which is phylogenetically distinct.

To assess whether litchi 24-nt phasiRNA exhibit as tight spatiotemporal control as grasses, we collected stages from early buds to fully open flowers. The dioecious flowers were segregated when the sex was identifiable (Fig. 2a), yielding five stages before sexual differentiation (MDS_I to MDS_V), male flowers from four stages (MFB_I to MFB_IV), and female flowers from three stages (FFB_I to FFB_III). The 24-nt phasiRNA abundance correlated with development of flower buds until a peak at stage MDS_V (Fig. 2a). After sexual specification, anthers either develop rapidly or abort; both miR2275 and 24-nt phasiRNAs were enriched in immature anthers, with slightly distinct peaks (Fig. 2a). In situ localization showed an enrichment of both miR2275 and 24-nt phasiRNAs in the tapetum and meiocytes of meiotic anthers, in which the archesporial cells and pollen become less arranged compared to cells in premeiotic stages (Fig. 2b). The spatiotemporal distribution in litchi was consistent with maize[4], suggesting a role in pollen development[8].

**Confirmation of 24-nt phasiRNA pathway in four eudicots.** The ephemeral abundance peak of reproductive 24-nt phasiRNAs

may explain why these highly abundant sRNAs were not previously discovered in eudicots. To ascertain their distribution in flowering plants, we examined more eudicots. *Citrus sinensis* (orange) is phylogenetically close to litchi; genomic analysis identified ten *MIR2275* precursors, with six mature miR2275 copies expressed (Supplementary Fig. 2A) plus 32 24-PHAS loci enriched in flowers (Supplementary Fig. 2A and Supplementary Data 2). The grape genome encodes ten *MIR2275* stem–loops (five unique mature miR2275 sequences detected), and inflorescence-enriched 24-nt phasiRNAs from 42 loci were identified (Fig. 3a, Supplementary Data 3). The strawberry genome (*Fragaria vesca*) encodes 17 *MIR2275* precursors (12 unique miR2275 sequences), but we previously failed to find miR2275 [11], likely reflecting utilization of uninformative floral staging. We collected early floral buds of strawberry (stages 6–9), and found both miR2275 and 24-nt phasiRNAs from 221 loci (Fig. 3b, Supplementary Data 4). We also found miR2275 and 24-nt phasiRNAs in cotton enriched in anthers at the tetrad stage (Supplementary Fig. 2B, Supplementary Data 5).

Conserved from these five eudicots to grasses is the clustering of *MIR2275* stem–loops, separated by just tens of nucleotides on a single precursor transcript (Supplementary Fig. 2C), an arrangement not described as conserved for other plant miRNAs. In litchi, three or two *MIR2275* stem–loops cluster (Fig. 1a, Supplementary Fig. 1A). In strawberry, as many as six *MIRNA* stem–loops cluster within ~1 kb, encoding four unique miR2275 sequences (Fig. 3c). We infer by its deep conservation that this unique, polycistronic precursor structure is of functional importance.

**Broad presence of the miR2275 phasiRNA pathway in eudicots.** As supported by extensive public sRNA data and genomic analyses, miR2275 is absent in Arabidopsis and in well-studied species in the Fabaceae and Solanaceae, nor have 24-nt phasiRNAs been described in these species. This discrepancy with the eudicots analyzed in the previous section supports variable presence/absence of the pathway across eudicots. We next assessed the presence of the pathway more broadly in eudicots, via in silico analysis of sequenced eudicot genomes. Given the tight association of miR2275 with 24-nt phasiRNAs, we analyzed 209 plant genomes for *MIR2275* loci. *MIR2275* was identified in the basal angiosperm *Amborella trichopoda* (Supplementary Fig. 2D, Supplementary Data 6), but absent from gymnosperms or earlier-diverging species (Supplementary Data 7). In monocots, *MIR2275* was in all grasses (Poales) plus close lineages (Asparagales, Arecales, and Commelinales), although absent in some basal monocots (Alismatales and Pandenales) (Fig. 3d, Supplementary Data 7). However, *MIR2275* was widely observed in eudicots (Fig. 3d, Supplementary Data 6), present in the Malpighiales, Fagales, Sapindales, Ericales, but absent in the Brassicales, Caryophyllales, Cucurbitaceae, and Lamiales. *MIR2275* was not detected in several well-studied plant families, including the Brassicaceae, Solanaceae, Cucurbitaceae, and Fabaceae. In almost all species containing *MIR2275*, at least one polycistronic *MIR2275* cluster was observed (Fig. 3d, Supplementary Data 7).

In grasses, only the specialized DCL5 generates 24-nt phasiRNAs[8]. Which DCL performs this role in eudicots? We searched for *DCL* genes from the 209 plant genomes. *DCL5* was monophyletic in monocots, suggesting another DCL generates 24-nt phasiRNAs in eudicots (Supplementary Fig. 3A). DCL3 is the only eudicot Dicer known to produce 24-mers, i.e. the 24-nt heterochromatic siRNAs (hc-siRNAs); DCL3 can process mRNA substrates, as demonstrated from tasiRNA loci in the Arabidopsis *dcl2/4* double mutant[12]. DCL3 is conserved in all eudicot species we examined (Supplementary Fig. 3A). Many

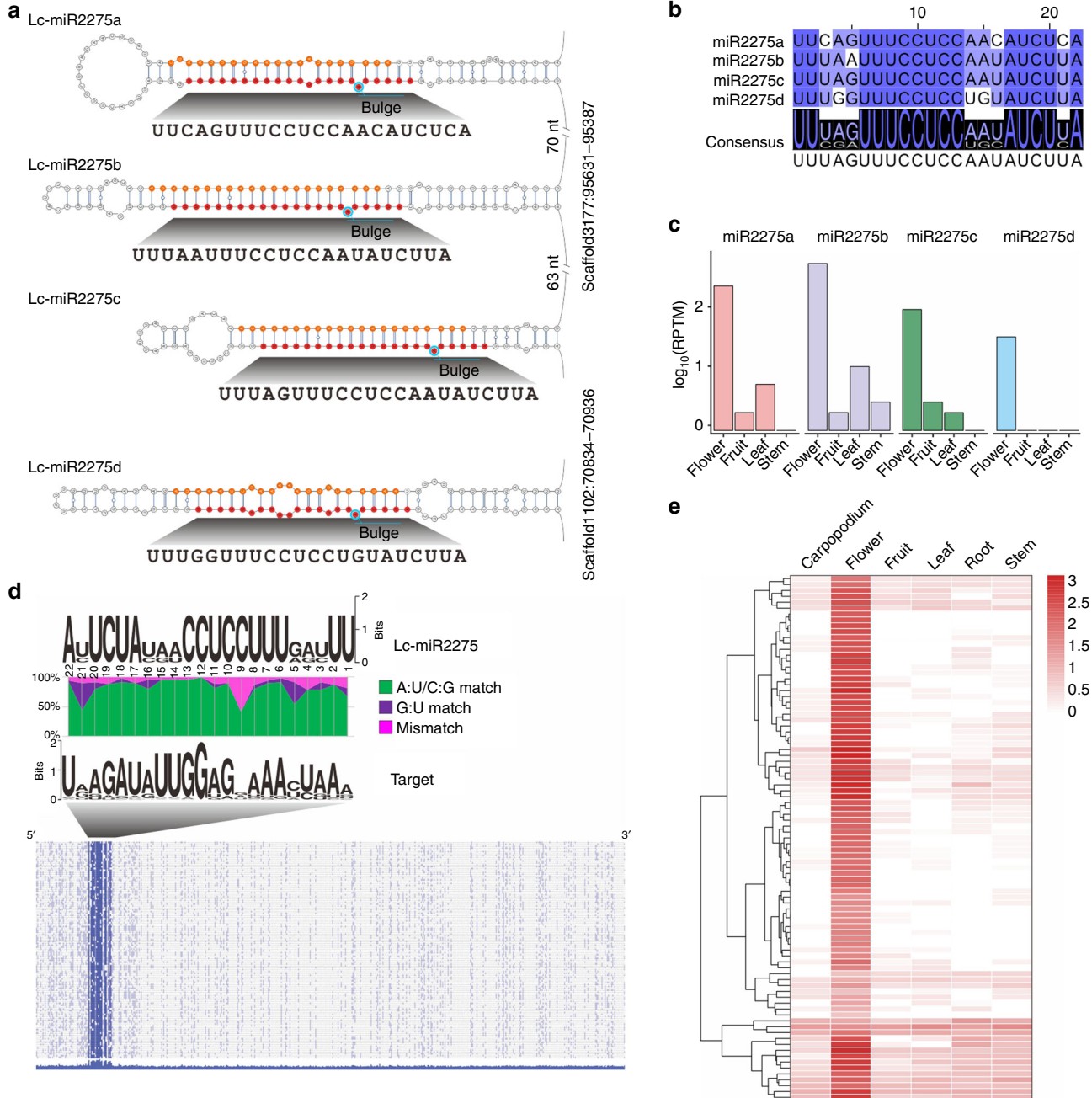

**Fig. 1** miR2275-triggered 24-nt reproductive phasiRNAs are abundant in litchi. **a** Stem–loop regions of two *MIR2275* precursors in litchi. miR2275 and miR2275* sequences are marked in red and orange, respectively. Bulges within the miR2275/miR2275* duplexes are indicated by a cyan circle. **b** Alignment of all expressed members of the miR2275 family in litchi. The degree of conservation for each nucleotide is represented by the color, with a dark color denoting a high level of conservation and a light color denoting a low level. The consensus sequence of the alignment is displayed below with sequence logos. **c** Expression of miR2275 members in four litchi tissues. **d** Above: alignment of miR2275 duplex and its target sites in 24-*PHAS* loci in litchi. The nucleotide pairing at each position between miR2275 and miR2275* is indicated by different colors, with A:U/C:G matches denoted in green, G:U matches in purple, and all mismatches in pink. Below: nucleotide sequence alignment of 24-*PHAS* loci with sequence similarity denoted by the density of blue color, demonstrating no conservation outside of the miR2275 target site (as in grass 24-*PHAS* loci). **e** Accumulation of 24-nt phasiRNAs in different tissues of litchi, with each row denoting a 24-*PHAS* locus, and each column a tissue as indicated. The key at right indicates the abundance in units of log2 (RPTM)

eudicot lineages demonstrated duplicated *DCL3* genes, yet just a few lineages had ancestral duplications, for instance, the Rosaceae (Supplementary Fig. 3B). However, *DCL3* is single copy in some eudicots with 24-nt phasiRNAs, for example, grape (*Vitis vinifera*) and rubber (*Hevea brasiliensis*) (Supplementary Fig. 3B). Therefore, we hypothesize that DCL3 in some eudicots has dual functions, processing distinct substrates

(RNA Pol II vs Pol IV) for biogenesis of either 24-nt phasiRNAs or 24-nt hc-siRNAs, a duality relieved in many monocots via subfunctionalization of DCL3 and DCL5. Application of machine learning demonstrated that angiosperm 24-nt phasiRNAs are more similar across species than to 24-nt hc-siRNAs from within a species (Supplementary Fig. 4), supporting the distinction of these pathways.

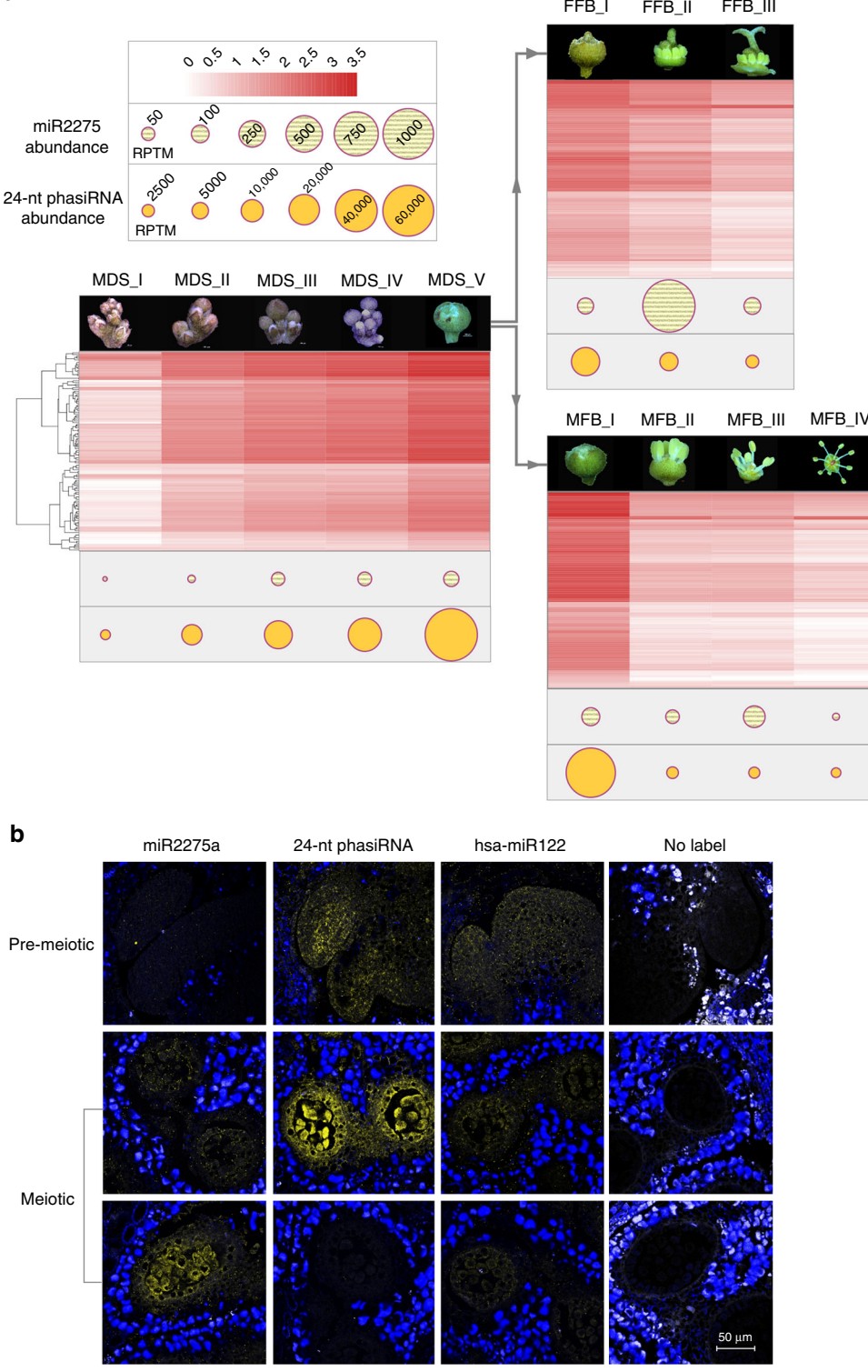

**Fig. 2** Spatiotemporal patterns of accumulation of miR2275 and 24-nt phasiRNAs in litchi. **a** Abundances of miR2275 and 24-nt phasiRNAs as indicated in the key (upper left) across developmental stages of litchi flowers. The heat map corresponds to the abundance of phasiRNAs at each 24-*PHAS* locus and dots indicate the summed abundances as indicated. MDS morphological differentiation stage, FFB female flower buds, MFB male flower buds. **b** In situ hybridization of miR2275 and a representative 24-nt phasiRNA in anthers from litchi flower buds. hsa-miR122 is a human miRNA used as a negative control, while the "no label" images have no labeled probe and serve as another negative control. Density of yellow denotes the sRNA hybridization signal; blue is autofluorescence of tissues surrounding the anthers. The diameter for litchi anthers increases as it develops from premeiotic to meiotic stage. At the premeiotic stage, the stamens are less than 1000 μm in diameter. Scale bar = 50 μm for all images

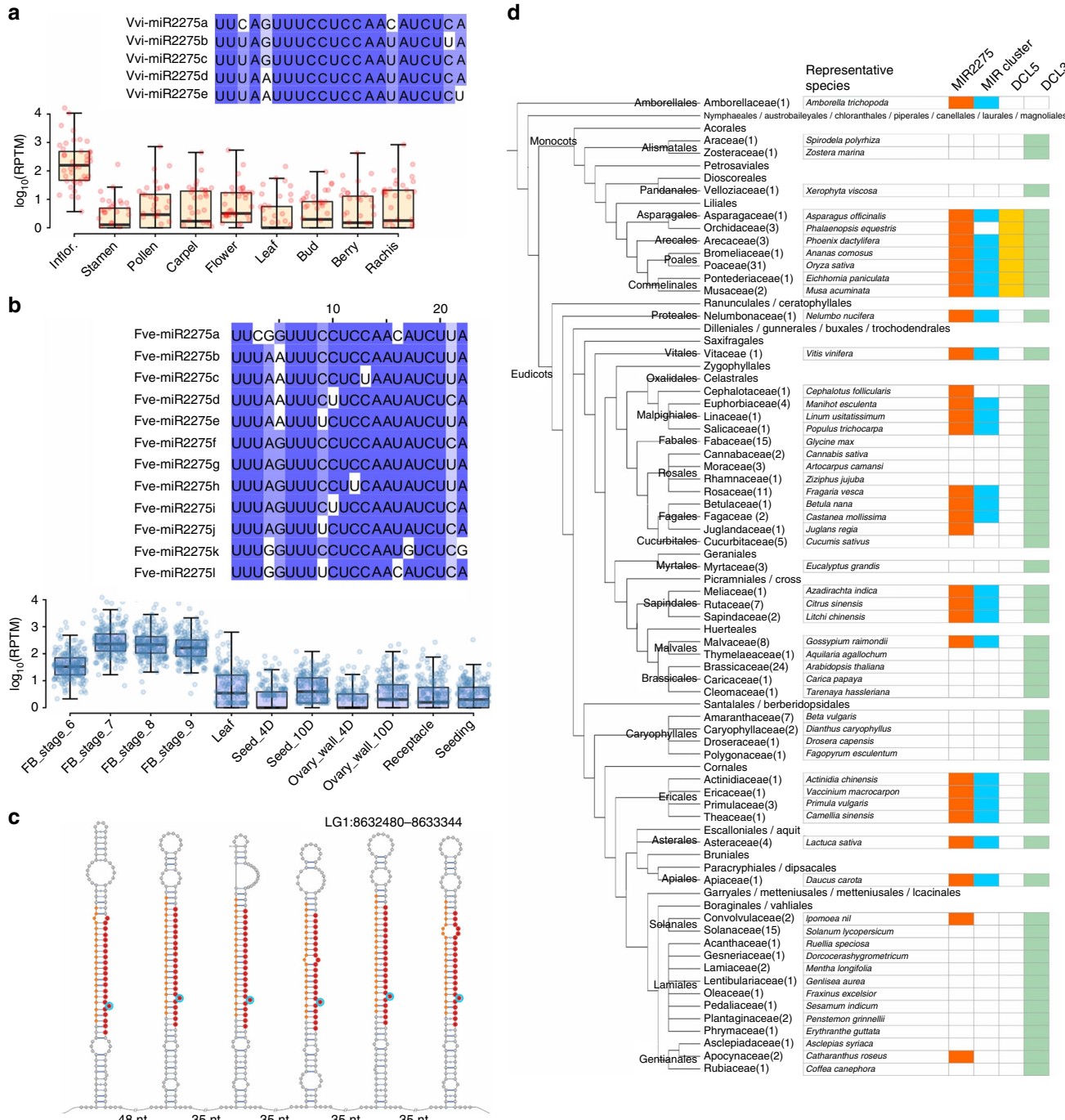

**Fig. 3** Wide conservation of the miR2275/24-nt phasiRNA pathway in eudicots. **a** miR2275 diversity and 24-nt phasiRNA accumulation in grape. Above: the degree of sequence conservation along the miRNAs is represented by the density of blue color. Below: summed abundances of sRNAs at each of 42 24-*PHAS* loci in reads per ten million (RPTM). In the boxplot, the center line represents the median; box limits are the upper and lower quartiles; whiskers are the 1.5× interquartile range; points show the scatter of data points. **b** miR2275 diversity and 24-nt phasiRNA accumulation in strawberry, as in panel **a**, but for 221 24-*PHAS* loci. The boxplot is drawn as in panel **a**. **c** A polycistronic precursor encoding six *MIR2275* stem–loops in strawberry, with miR2275 and miR2275* marked in red and orange, respectively. Bulges within the miR2275/miR2275* duplexes are indicated by a cyan circle. **d** A tree illustration of the conservation of the miR2275/24-nt phasiRNA pathway in plants. The plant phylogeny (left) was constructed according to the NCBI Taxonomy Common Tree. Plant orders or families denote representative plant species in a given order or family with genome sequenced and encoding miR2275 in the genome (black), with genome sequence but no miR2275 found (blue), and no genome sequenced (light gray). Numbers in parentheses after the family name indicate the number of plant species of that family used in this study. The presence of a given element (miR2275, *DCL5*, *DCL3*) or feature (*MIR2275* in a polycistronic cluster) is indicated by colored boxes

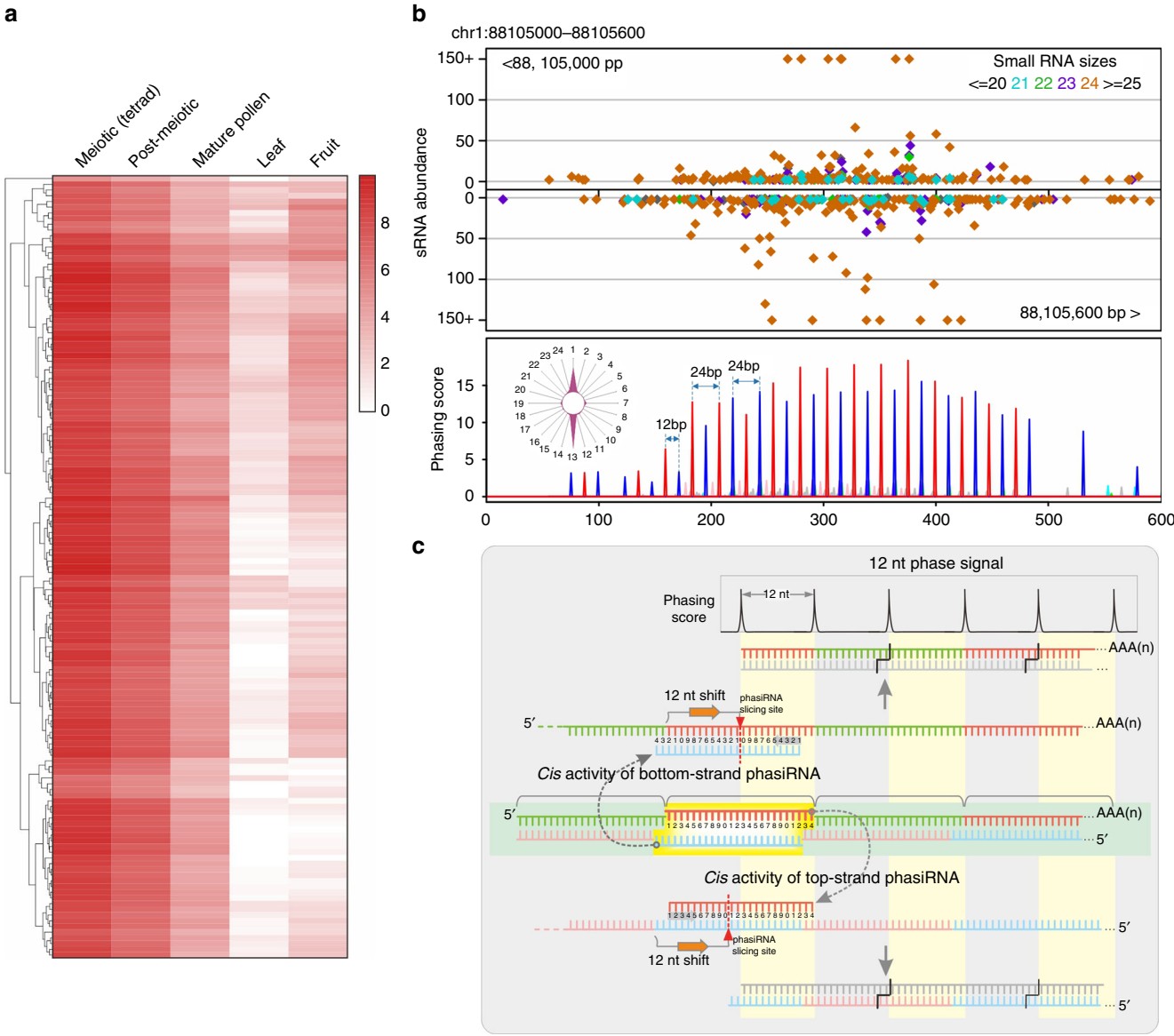

**Fig. 4** 24-nt phasiRNAs with 12-nt phasing in tomato. **a** 24-nt phasiRNA accumulation in different tissues of tomato, with each row denoting a 24-*PHAS* locus. **b** A representative locus generating 24-nt phasiRNAs in tomato. Distribution of sRNAs (above) and deduced phasing score (below) are viewed along the coordinates of the *PHAS* locus, with phasiRNA distribution in registers displayed in a radar plot on the left. **c** A hypothesis for the generation of 24-nt phasiRNAs with a 12-nt phasing pattern. *Cis* activity of either strand of a 24-nt phasiRNA duplex could trigger secondary phasing with a 12-nt shift to the primary 24-nt phase. The combination of the primary and secondary phases leads to the formation of the 12-nt phase pattern

**24-nt phasiRNAs with distinct features in tomato and petunia.** An analysis of data from Solanaceous species unexpectedly (because of an absence of miR2275) yielded 137 highly confident 24-*PHAS* loci enriched in meiotic tomato inflorescences (Fig. 4a, Supplementary Data 8). These loci had unique characteristics: no apparent miRNA target site and a 12-nt phasing of the 24-nt siRNAs (Fig. 4b). Similar loci were observed in petunia (Supplementary Fig. 5, Supplementary Data 9 and 10). We revisited the 24-*PHAS* loci of rice and maize and found a number of 12-nt phased loci also lacking an miR2275 target site (Supplementary Fig. 6); in the absence of the miR2275 target site, phasing was less precise (Supplementary Fig. 6). How is 12-nt phasing generated? One possibility is *cis* activity of 24-nt phasiRNAs, demonstrated for 24-nt heterochromatin siRNAs[13], but in this case *cis*-directed cleavage from either source strand would promote the generation of secondary 24-nt phasiRNAs shifted by 12 nt (Fig. 4c,

Supplementary Fig. 7). Yet, it is unclear how this feedback loop of *cis*-triggered phasiRNA biogenesis would initiate—a topic for future analyses.

## Discussion

We conclude that 24-nt phasiRNAs emerged in angiosperms, but were lost in some lineages, presumably reflecting life history adaptations. Genetic components of pathway were enhanced in monocots via specialization of DCL5. If 24-*PHAS* loci predate the DCL3/DCL5 duplication and divergence, the emergence of DCL5 could represent a case of "escape from adaptive conflict" (EAC), the theory that gene duplication can segregate functions from a single-copy progenitor[14]. EAC was invoked as an explanation for the expansion of RNA polymerases in plants[15], including Pol IV subunits that make DCL3 substrates for hc-siRNAs. Coupled with

the demonstration that 24-nt phasiRNAs are required for full male fertility in maize[8], our observations extend the analogy of the independently evolved plant reproductive phasiRNAs and metazoan piRNAs, including our observation that in some plant lineages an alternative mechanism of biogenesis may occur, that is, *cis*-mediated cleavage that is reminiscent of the "ping-pong" production of piRNAs[16].

The presence of the miR2275-24–*PHAS*–phasiRNA pathway across the angiosperms supports an important functional role for these molecules, likely in reproduction, as recently demonstrated[8]. However, the absence of miR2275 in many eudicots and some basal monocots suggests that the function of this pathway is dispensable for the development of some plant species. Alternatively, perhaps the 24-nt phasiRNAs are indispensable in angiosperms, but they are produced without miR2275, as found in tomato and petunia, and in such a spatiotemporally limited manner that they have escaped detection in the Brassicaeae, Fabaceae, etc. Yet another possibility is that there are other pathways in these lineages that functionally substitute for the role of the miR2275-24–*PHAS*–phasiRNA pathway, without the necessity of generating phasiRNAs. As with mammalian meiotic piRNAs, the functions of plant meiotic phasiRNAs are as yet largely unclear; as these are worked out in the coming years, the answers to these questions are likely to be addressed.

## Methods

**Plant materials**. Early-stage flower buds for litchi (Supplementary Fig. 8) were collected from nine litchi trees, cultivar "Feizixiao", grown together in an orchard at South China Agricultural University (Guangzhou, China). Samples from three sets of three trees per biological replicate were used for sequencing. Flower buds of 12 different stages with three biological replicates were collected, yielding 36 samples in total. Strawberry (*Fragaria vesca*) inbred line Yellow Wonder 5AF7 (YW5AF7) was grown in a growth chamber with 12 h light at 25 °C followed by 12 h dark at 20 °C. Anthers from different stages including buds from stages 6 to 9[17] were hand-dissected under a stereomicroscope and frozen immediately in liquid nitrogen.

**sRNA sequencing**. For flower buds of litchi, total RNA was extracted using TRIzol (Thermo Fisher Scientific) plus Fruit-mate (Takara) for polysaccharide/polyphenol removal, both according to the manufacturers' instructions. For sRNA sequencing, 10 μg of the mixed total RNAs with RNA integrity number (RIN) ≥ 7.5 were used for sRNA library construction. sRNA libraries were constructed and sequenced on Illumina HiSeq 2500 platform at RIBOBIO (Guangzhou, China).

RNA of strawberry flower buds was extracted using PureLink Plant RNA Reagent. sRNA libraries were constructed using the Illumina TruSeq sRNA kit, and sequenced on the Illumina HiSeq platform at the University of Delaware.

**sRNA data analysis and *PHAS* locus annotation**. Annotation of *PHAS* loci was conducted largely by application of a *P*-value-based approach[18]. The *P*-value cutoff was set to 0.001. In the initial analyses (as described in the main text), only genomic loci with a target site of miR2275 were considered as valid *PHAS* loci and used for further analyses.

**Phylogenetic tree construction**. DCL protein sequences were identified from each plant genome using genBlastG[19] and aligned using MUSCLE[20] and then trimmed using trimAL[21]. The resulting multiple alignment was utilized to construct the phylogenetic tree using RAxML[22].

**In situ RNA hybridizations**. sRNAs were detected using LNA probes by Exiqon (Woburn, MA). Samples were vacuum fixed using 4% paraformaldehyde, and submitted to a histology lab (A.I. DuPont Hospital for Children) for paraffin embedding. We followed published protocols for the prehybridization, hybridization, and post-hybridization steps[23]. For the detection step, fluorescent in situ hybridization was carried out using anti-Digoxigenin Fab fragment (Sigma-Aldrich cat# 11214667001) and donkey anti-sheep IgG (H + L) AF647 (Thermo Fisher Scientific cat# A-21448) antibody combination. We used a Zeiss LSM 880 multi-photon confocal microscope for the final imaging process with an excitation of 745 nm and EC Plan-Neofluar 40×/1.30 oil DIC objective (University of Delaware Bioimaging Center). Probe sequences are listed in Supplementary Data 11.

**Reporting Summary**. Further information on experimental design is available in the Nature Research Reporting Summary linked to this article.

## Data availability

All sRNA data generated from this study were deposited in NCBI SRA (Sequence Read Archive) under the accession number SRP149511 (for litchi) [https://www.ncbi.nlm.nih.gov/sra/?term=SRP149511] and SRP149613 (for strawberry) [https://www.ncbi.nlm.nih.gov/sra/?term=SRP149613]. sRNA data from public databases used for our analysis are all listed in Supplementary Data 12.

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

## Acknowledgements

This work was funded by the National Key Research and Developmental Program of China (2018YFD1000104). This work was also supported by awards to R.X. or J.L. from the National Natural Science Foundation of China (#31872063, #31471859), the Outstanding Talent Program of the Ministry of Agriculture, the Guangzhou Science

and Technology Key Project (201804020063), and the Innovation Team Project of the Department of Education of Guangdong Province (2016KCXTD011), and an award from the US National Science Foundation Plant Genome Research Program (#1339229) to B.C.M. Microscopy equipment was acquired with a shared instrumentation grant (S10 OD016361) and access was supported by the NIH-NIGMS (P20 GM103446), the NSF (IIA-1301765), and the State of Delaware, USA.

## Author contributions

R.X., J.L., and B.C.M. conceived the study and designed all the analyses. R.X. and C.C. performed most of the data analyses. W.M., S.P., and J.X. collected samples or constructed sRNA libraries. F.W. and Z.L. collected strawberry tissues. K.H performed in situ hybridizations. P.P. helped with the nucleotide composition analysis of phasiRNAs using machine learning approach. R.X., K.H., and C.C. prepared the figures. R.X. and B.C.M wrote the paper.

## Additional information

**Competing interests:** The authors declare no competing interests.

