## [Peer Review File · Nature Communications]

Reviewers' comments:

Reviewer #1 (Remarks to the Author):

The paper demonstrated the presence of the reproduction-specific 24-nt phasiRNAs among several plant species. The study showed that miR2275 and 24-nt phasiRNAs are expressed in the eudicot litchi and other dicot plants, which proved that phasiRNAs are not restricted to monocots. Interestingly, a wider analysis of plants genomes demonstrated that only a branch of monocots possesses a full set of the known "phasiRNA" players described in rice and maize, whereas other monocots and eudicots have either a partial set or no component at all like Arabidopsis. This raises an interesting question about the impact of evolution on miR2275, 24-nt phasiRNA and DCL3/5 presence, and their biological functions. Since phasiRNAs are important for gametogenesis in rice and maize, this study suggested the importance of this conserved pathway in plant reproduction. Although no molecular mechanism was involved in this study, this discovery is very interesting and important. In general, this manuscript is well written and the amount of work presented in the paper is appreciated.

A few minor comments are listed below.

1. In abstract, "two pathways yield abundant phasiRNA during anther development." I anticipate to learn what two pathways are in the next sentence. If it is due to the length limit, perhaps rewording this sentence- "two pathways yield abundant phasiRNA during anther development."
2. Figure 2A: It is difficult for readers to understand how the authors differentiate the different stages of flower by the small photo attached especially stages before sexual differentiation. Perhaps the authors can give a brief description on all the developmental stages of flower according to figures of "five stages before sexual differentiation (MDS_I to MDS_V), male flowers from four stages (MFB_I to MFB_IV), and female flowers from three stages (FFB_I to FFB_III)" in supplemental materials.
3. Figure 2B, it is confusing to have "negative control" referred to both human miRNA probes and no labeled probe.
4. Figure 2B: The authors showed that in situ localization of miR2775 and 24-nt phasiRNAs was enriched in tapetum and meiocytes. The size of anthers used in this experiment was not mentioned in the Material and Methods. Brief descriptions on appearance of premeiotic and meiotic stages in figures should also be added in the text.
5. What is the role of miR2275 accumulation in FFB_II, since it occurs after accumulation of 24nt-phasiRNAs? This also is seen in Fig 2B by in situ hybridization?
6. The authors mentioned that MIR2775 is absent from gymnosperms or earlier diverging species. How many genomes of these species has been checked?
7. Figure 3D: it is difficult to read the figure. The light grey and blue colors not clear and not distinct enough from others.

Reviewer #2 (Remarks to the Author):

This wide examination of special reproductive small RNAs, thought to be limited to monocot grasses, concisely reveals that "24-nt reproductive phasiRNAs are broadly present in angiosperms". This is an exciting surprise, and relevant to all plant researchers working on reproduction and/or small RNAs.

The authors generated new sequencing data from litchi and strawberry, and also made laudable use of publically available data which they thoroughly mined. The text is well written, the figures informative and clear, and supplemental data conforms in style and value.

Other than few minor requests for e.g. addition of descriptions for abbreviations, scales and figure legend (see attached files), I do not have any other concerns or improvement suggestions.

Only one main open question remains (but cannot be answered by the present study though readers will wonder at the end, and thus the authors should maybe speculate about): the functional mechanistic of phasiRNAs – what could the function be? Other than meiosis-specific DNA methylation and speculative roles for meiotic chromosome behavior (see Dukowic-Schulze et al 2016), no function has been shown. Interesting in view of the recent data is the question why some lineages seem to have/ need/ retain phasiRNAs why others can cope without.

Reviewer #3 (Remarks to the Author):

Nature Communications
To Authors,

This paper showed that 24-nt reproductive phasiRNAs are specifically expressed during reproduction in litchi, strawberry and tomato, and in silico analysis revealed that miR2275, which may trigger 24-nt phasiRNAs production, are conserved in angiosperms, not only Poaceae family. Recently, the authors' group has reported about 24-nt reproductive phasiRNAs/miR2275/DCL5 (DCL3b) in nongrass monocots (Kakrana et al., 2018), so the new information defined in this study over previous work is rather limited. Furthermore, the overall picture of 21-nt or/and 24-nt phasiRNAs function thus remains unclear. More molecular experimental evidence should be shown in this journal to prove the importance of 24-nt reproductive phasiRNAs. It recommends to perform additive experiments that are discussed below about PHAS/phasiRNAs/DCLs.

Major comments

1) 21-nt reproductive phasiRNAs in dicots

Some groups have revealed that 21-nt phasiRNAs are present in angiosperm and gymnosperm. How about 21-nt REPRODUCTIVE phasiRNAs in dicots, though 21-nt phasiRNAs are mainly derived from defense genes (NB-LRRs) in leguminous? At least, please show whether there are 21-nt reproductive phasiRNAs in litchi chinese, orange, grape, (strawberry) and tomato.

2) miR2118 (in silico analysis)

Please add the information whether miR2118 exist or not in tree illustration of Figure3D, similar to miR2275 (in 209 genome).

3) 21nt/24nt reproductive phasiRNAs in angiosperm

Please add the information of 21-nt reproductive phasiRNAs (or 21-PHAS) and 24-nt reproductive phasiRNAs (or 24-PHAS) including all of references in this paper (ex Figure 3D or other) to understand reproductive phasiRNAs more deeply and broadly, because the authors have performed identification of 24-nt phasiRNAs using just 3-5 species in dicots in this paper. The subtitle "A tree illustration of the conversation of the miR2275/24-nt phasiRNA pathway" is written in figure 3D legend. But there are no 24-nt phasiRNAs information in Figure 3D. Please add the information of 24nt-reproductive phasiRNAs (or 24 PHAS) too.

Moreover, if 21-nt reproductive phasiRNAs are broadly exist in plants, please explain the relationship between 21-nt and 24-nt reproductive phasiRNAs in angiosperm. Premeiotic or meiotic in angiosperm etc.

4) 24-PHAS in dicots.

24-PHASs in grass mostly locate at intergenic regions, and 24-nt phasiRNAs of nongrass are derived from inverted repeat. Please show the annotation (or feature) of 24-PHAS in each species identified in this work.

5) DCL5 (Sup Figure4)

Recently, The Meyers group starts to use DCL5 as a monophyletic DCL in monocots, though OsDCL5 is OsDCL3b. SupFig4 A is ambiguous to detect the duplication that gave rise to the DCL5 clade. Please show the high resolution tree of SupFig4B including Monocot DCL5 (red) and DCL3 (yellow) parts, and mark the duplication that gave rise to DCL5 clade and lineage-specific duplications in this tree.

6) More molecular experimental evidence

I mentioned above that understanding of 24-nt phasiRNAs deepens more by molecular experimental evidence. Please add the following experiments or another. (For exam1. Producing/analyzing the DCL3 mutant in the dicot, in which 24-nt reproductive phasiRNAs are expressed. Forexam2. Including the function of maize DCL5 (Zhang et al) to this paper.)

7) SupFigure 2 and SupTable 2

In manuscript (page 4), the authors show 36 24-PHAS in citrus, although there are just 32 24-PHAS in SupTable2. Furthermore, it seems that phasiRNAs are not expressed at about half of PHAS of both GSM455228 and GSM455230 in SupTable2. There are no information about GSM455228, GSM455230 and Orange_flower_SRR847766_1. Please explain more detail in SupFigure2 A and B.

8) Machine learning

It was reported that most of 5'terminal nucleotide of 24-nt hcsiRNAs are "A" (Mi et al., 2008). In this machine learning (SupFig5), it is difficult to detect the difference between phasiRNA and hcsiRNA except for 5'terminal nucleotide. Please show likely score or something to understand the difference of small RNAs feature.

9) Negative control (ISH)

Please show the LNA sites of all probes including has-miR122 (in SupTable10).

10) It is recommended to fit this manuscript to Nature communications form.

We would like to thank the reviewers for their comments and suggestions. Below we have responded to the reviewers' comments one by one. Our comments, like this one, are preceded by ">>>" and in blue fonts.

Reviewers' comments:

Reviewer #1 (Remarks to the Author):

The paper demonstrated the presence of the reproduction-specific 24-nt phasiRNAs among several plant species. The study showed that miR2275 and 24-nt phasiRNAs are expressed in the eudicot litchi and other dicot plants, which proved that phasiRNAs are not restricted to monocots. Interestingly, a wider analysis of plants genomes demonstrated that only a branch of monocots possesses a full set of the known "phasiRNA" players described in rice and maize, whereas other monocots and eudicots have either a partial set or no component at all like Arabidopsis. This raises an interesting question about the impact of evolution on miR2275, 24-nt phasiRNA and DCL3/5 presence, and their biological functions. Since phasiRNAs are important for gametogenesis in rice and maize, this study suggested the importance of this conserved pathway in plant reproduction. Although no molecular mechanism was involved in this study, this discovery is very interesting and important. In general, this manuscript is well written and the amount of work presented in the paper is appreciated.

>>> *We appreciate the positive comments.*

A few minor comments are listed below.

1. In abstract, "two pathways yield abundant phasiRNA during anther development." I anticipate to learn what two pathways are in the next sentence. If it is due to the length limit, perhaps rewording this sentence- "two pathways yield abundant phasiRNA during anther development."

>>> *Fixed, thanks.*

2. Figure 2A: It is difficult for readers to understand how the authors differentiate the different stages of flower by the small photo attached especially stages before sexual differentiation. Perhaps the authors can give a brief description on all the developmental stages of flower according to figures of "five stages before sexual differentiation (MDS_I to MDS_V), male flowers from four stages (MFB_I to MFB_IV), and female flowers from three stages (FFB_I to FFB_III)" in supplemental materials.

>>> *Thanks. We have added a supplemental figure (Figure S9) to give more information about the staging of litchi flowers.*

3. Figure 2B, it is confusing to have "negative control" referred to both human miRNA probes and no labeled probe.

>>> *Fixed, thanks. We usually have two controls for each in situ hybridization experiment: one is a non-specific miRNA control (hsa-miR122 in this manuscript), and the other one is a "no label" control that is used to indicate the background level for imaging processing. While these are both negative controls, to minimize confusion, we have changed the wording for the figure legend from "negative control" to "No label".*

4. Figure 2B: The authors showed that in situ localization of miR2775 and 24-nt phasiRNAs was enriched

in tapetum and meiocytes. The size of anthers used in this experiment was not mentioned in the Material and Methods. Brief descriptions on appearance of premeiotic and meiotic stages in figures should also be added in the text.

>>> *Fixed, thanks. The diameter for litchi anthers increases as it develops from premeiotic to meiotic stage (as shown below). At the premeiotic stage, the stamens are less than 1000 μm in diameter. Cell layers, especially the tapetal layer, are not fully developed in premeiotic stages. The cells are aligned at this point, yet once the pollen mother cells develop, the archesporial cells and pollen become less arranged as can be seen from Figure 2B middle and bottom rows (meiotic stages), and as shown below. Brief descriptions were added in both the main text and the legend of figure 2.*

5. What is the role of miR2275 accumulation in FFB_II, since it occurs after accumulation of 24nt-phasRNAs? This also is seen in Fig 2B by in situ hybridization?

>>> *This is a good point and was also a question that we considered. A possible explanation is that this is simply functionally unimportant and inconsequential overaccumulation. In other words, although the 24-nt phasiRNA production requires the presence of both the miR2275 trigger and the 24-PHAS transcripts (and of course the components of the phasiRNA generation machinery, Dicer, Argonaute, RdRP, etc.), the 24-PHAS mRNAs are the primary determinant of phasiRNA production. miR2275 is necessary but not sufficient to determine phasiRNA abundances, so the abundance of miR2275 and an accumulation has no particular cost or benefit to the anther. In other words, the 24-PHAS transcripts are highly expressed at the stages of MDS_V, FFB_I, and MFB_I, in which the abundance of miR2275 is sufficient to cleave those transcripts and trigger phasiRNA production, therefore leading to the maximum level of phasiRNAs at those stages.*

6. The authors mentioned that MIR2775 is absent from gymnosperms or earlier diverging species. How many genomes of these species has been checked?

>>> *We checked a liverwort genome (*Marchantia polymorpha*), as well as genomes from a moss (*Physcomitrella patens*) and five gymnosperms (*Picea abies*, *Picea glauca*, *Pinus lambertiana*, *Pinus sylvestris*, *Pinus taeda*).*

7. Figure 3D: it is difficult to read the figure. The light grey and blue colors not clear and not distinct

enough from others.

>>> *Thanks. We have removed the color code.*

Reviewer #2 (Remarks to the Author):

This wide examination of special reproductive small RNAs, thought to be limited to monocot grasses, concisely reveals that “24-nt reproductive phasiRNAs are broadly present in angiosperms”. This is an exciting surprise, and relevant to all plant researchers working on reproduction and/or small RNAs. The authors generated new sequencing data from litchi and strawberry, and also made laudable use of publically available data which they thoroughly mined. The text is well written, the figures informative and clear, and supplemental data conforms in style and value.

>>> *Thanks, we appreciate the positive comments.*

Other than few minor requests for e.g. addition of descriptions for abbreviations, scales and figure legend (see attached files), I do not have any other concerns or improvement suggestions.

Only one main open question remains (but cannot be answered by the present study though readers will wonder at the end, and thus the authors should maybe speculate about): the functional mechanistic of phasiRNAs – what could the function be? Other than meiosis-specific DNA methylation and speculative roles for meiotic chromosome behavior (see Dukowic-Schulze et al 2016), no function has been shown. Interesting in view of the recent data is the question why some lineages seem to have/ need/ retain phasiRNAs why others can cope without.

>>> *Indeed, these are major questions for us too. As there are many 24-PHAS loci in a single species, it is difficult to manipulate all of them simultaneously, so we are planning to knock out the MIR2275 genes using CRISPR in diploid strawberry. But this may not be easy to achieve as this is a multi-gene family in strawberry, and transformation is quite challenging. Regarding the lineage-specific presence of the pathway, we think that (1) this pathway is important but not indispensable, or (2) there are other alternatives of this pathway in some species, like what we found in tomato. Why this is the case is a fascinating question.*

Reviewer #3 (Remarks to the Author):

Nature Communications
To Authors,

This paper showed that 24-nt reproductive phasiRNAs are specifically expressed during reproduction in litchi, strawberry and tomato, and in silico analysis revealed that miR2275, which may trigger 24-nt phasiRNAs production, are conserved in angiosperms, not only Poaceae family. Recently, the authors' group has reported about 24-nt reproductive phasiRNAs/miR2275/DCL5 (DCL3b) in nongrass monocots (Kakrana et al., 2018), so the new information defined in this study over previous work is rather limited. Furthermore, the overall picture of 21-nt or/and 24-nt phasiRNAs function thus remains unclear. More molecular experimental evidence should be shown in this journal to prove the importance of 24-nt

reproductive phasiRNAs. It recommends to perform additive experiments that are discussed below about PHAS/phasiRNAs/DCLs.

>>> We disagree that the new information here is limited. There is a major difference between a pathway that is limited to the monocots and one that is broadly found in the angiosperms, and it changes our perspective completely about the functional importance of these small RNAs. Indeed, the demonstration that the pathway emerged prior to DCL5 and that this would necessarily require a dual function for DCL3 prior to the emergence of DCL5 – this is big news. Plus, the observation that miR2275 has a conserved tandem hairpin structure, and the observation that the Solanaceous species have evolved a miR2275-independent pathway for the production of the 24-nt reproductive phasiRNAs...to our way of thinking, these are strikingly novel and exciting observations.

Both here and below, this reviewer is somewhat fixated on the 21-nt reproductive phasiRNAs, which are not the point of the paper. It's as if we were describing the auxin receptor but the reviewer keeps asking about GA. The 21- and 24-nt reproductive phasiRNAs are genetically separable, possibly independently evolved pathways that function in somewhat distinct stages in anther development. The 21-mers are dependent on miR2118, a miRNA which we and others have described has an alternate role in targeting NLR transcripts, and thus the sort of analyses we performed here are not applicable to that pathway. We would have had to conduct an entirely different sort of analysis (like auxin vs GA, for example).

Major comments

1) 21-nt reproductive phasiRNAs in dicots

Some groups have revealed that 21-nt phasiRNAs are present in angiosperm and gymnosperm. How about 21-nt REPRODUCTIVE phasiRNAs in dicots, though 21-nt phasiRNAs are mainly derived from defense genes (NB-LRRs) in leguminous? At least, please show whether there are 21-nt reproductive phasiRNAs in litchi chinease, orange, grape, (strawberry) and tomato.

>>> We appreciate the suggestion, and this would be a good topic for a future study, involving distinct types of analyses.

2) miR2118 (in silico analysis)

Please add the information whether miR2118 exist or not in tree illustration of Figure3D, similar to miR2275 (in 209 genome).

>>> This is an inappropriate analysis for this paper. We already know the miR482/2118 superfamily (including miR2118) is widely present in seed plants. We have published on the presence of miR482/2118 and potential 21-nt reproductive phasiRNAs in a gymnosperm (Norway spruce, doi: 10.1093/molbev/msv164), and we have demonstrated that the miRNA family has different target preferences between eudicots (mainly targeting NB-LRR genes) and monocots (predominantly targeting non-coding 21-PHAS loci). Based on our analysis, focusing on litchi miRNAs and 21-nt phasiRNAs (doi: 10.1111/nph.14934), the miR482/2118 family predominantly targets NB-LRRs in this species, and doesn't have as many non-coding targets as in monocots.

3) 21nt/24nt reproductive phasiRNAs in angiosperm

Please add the information of 21-nt reproductive phasiRNAs (or 21-PHAS) and 24-nt reproductive phasiRNAs (or 24-PHAS) including all of references in this paper (ex Figure 3D or other) to understand

reproductive phasiRNAs more deeply and broadly, because the authors have performed identification of 24-nt phasiRNAs using just 3-5 species in dicots in this paper.

>>> *Again, this paper is focused on the novel observations of the miR2275 pathway and its evolution and emergence.*

The subtitle “A tree illustration of the conversation of the miR2275/24-nt phasiRNA pathway” is written in figure 3D legend. But there are no 24-nt phasiRNAs information in Figure 3D. Please add the information of 24nt-reproductive phasiRNAs (or 24 PHAS) too.

>>> *Thanks. Sorry for the confusion. Actually we don't have the information on the 24-nt phasiRNA-generating loci because most of the species we check do not have sRNA data or data from the right stage. We validated the presence of 24-nt phasiRNA in litchi, orange, strawberry, grape and cotton – species for which we have good genomes and well-staged anther data. In other cases, we used miR2275 as a marker for the pathway. We have corrected the legend to be more specific.*

Moreover, if 21-nt reproductive phasiRNAs are broadly exist in plants, please explain the relationship between 21-nt and 24-nt reproductive phasiRNAs in angiosperm. Premeiotic or meiotic in angiosperm etc.

>>> *They are genetically separable pathways. Current evidence suggests that they emerged independently and evolved separately.*

4) 24-PHAS in dicots. 24-PHASs in grass mostly locate at intergenic regions, and 24-nt phasiRNAs of nongrass are derived from inverted repeat. Please show the annotation (or feature) of 24-PHAS in each species identified in this work.

>>> *We have checked the genomic location of 24-PHASs we identified in a few species (as shown in the figure below. Similar to monocots, they are mostly distributed in intergenic regions.*

5) DCL5 (Sup Figure4)

Recently, The Meyers group starts to use DCL5 as a monophyletic DCL in monocots, though OsDCL5 is OsDCL3b. SupFig4 A is ambiguous to detect the duplication that gave rise to the DCL5 clade. Please show

the high resolution tree of SupFig4B including Monocot DCL5 (red) and DCL3 (yellow) parts, and mark the duplication that gave rise to DCL5 clade and lineage-specific duplications in this tree.

>>> We would argue that DCL5 used to be called DCL3b – which, as the reviewer points out, came from the “Os” (rice) gene identifier. This designation was assigned when only one monocot genome was sequenced, and before any gene functions were known. Now that we know so much more, and know that this is a functionally distinct Dicer, it makes sense to assign it a functionally distinct name. This is the stand process in science. As we demonstrate, some genomes do contain recent, lineage specific gene duplicates, including of DCL3. If these in the eudicots were then also named “DCL3a” and “DCL3b” and the monocot DCL3b were retained, but the functions were not equivalent, the field would be a confusing mess of poor nomenclature. It makes more sense to update names to reflect the current state of our understanding.

A full tree of the DCL3/5 family is provided for reviewing purpose as a separate PDF file. We can clearly see that the DCL5 clade is restricted to the monocots. We did not include the whole tree as a supplemental is because the tree is too big, and the emergence of DCL5 in monocots is not the focus of our story.

6) More molecular experimental evidence

I mentioned above that understanding of 24-nt phasiRNAs deepens more by molecular experimental evidence. Please add the following experiments or another.

(For exam1. Producing/analyzing the DCL3 mutant in the dicot, in which 24-nt reproductive phasiRNAs are expressed. Forexam2. Including the function of maize DCL5 (Zhang et al) to this paper.)

>>> Yes, thanks, we have added a more clear mention of our functional work of maize DCL5 to this paper. Mutating DCL3 in a eudicot that has the 24-nt reproductive phasiRNAs is more challenging, and is among our long-term aims. We did contact David Baulcombe about the tomato dcl3 mutant, but it has apparently not proven viable in his lab.

7) SupFigure 2 and SupTable 2

In manuscript (page 4), the authors show 36 24-PHAS in citrus, although there are just 32 24-PHAS in SupTable2. Furthermore, it seems that phasiRNAs are not expressed at about half of PHAS of both GSM455228 and GSM455230 in SupTable2. There are no information about GSM455228, GSM455230 and Orange_flower_SRR847766_1. Please explain more detail in SupFigure2 A and B.

>>> Thanks. We have corrected the number in the main text. Both datasets GSM455228 and GSM455230 are from orange fruit, not the tissue in which 24-nt phasiRNAs are most abundant. We use them and the dataset SRR847765 as negative control to compare with the SRR847766 (the early flower bud), confirming that the 24-nt phasiRNAs are enriched in reproductive tissues.

8) Machine learning

It was reported that most of 5'terminal nucleotide of 24-nt hcsiRNAs are “A” (Mi et al., 2008). In this machine learning (SupFig5), it is difficult to detect the difference between phasiRNA and hcsiRNA except for 5'terminal nucleotide. Please show likely score or something to understand the difference of small RNAs feature.

>>> Thanks. Yes, the 5' terminal nucleotide is more dramatically different between phasiRNAs and hcsiRNAs. Beside that position there are indeed other positions are significantly different, and there

position are marked by square boxes. In order to make them more obvious, we have highlighted those positions with red dotted circles.

9) Negative control (ISH)

Please show the LNA sites of all probes including has-miR122 (in SupTable10).

>>> We designed and ordered the probes from Exiqon. However, Exiqon does not release the LNA sites to their customers. We are sorry about this, but presumably reordering the probes of the same sequences would result in the same LNA positions.

10) It is recommended to fit this manuscript to Nature Communications form.

>>> Fixed, thank you.